# MSAVQ: Multi-dimensional Sensitivity-Aware Vector Quantization for VLMs

## Abstract

Vision-Language Models (VLMs) have achieved remarkable progress, but their massive scale severely limits deployment in resource-constrained settings. Among existing compression strategies, vector quantization (VQ) stands out for its strong representational power under ultra-low bitwidths. VQ achieves this by constructing a compact codebook, where weight vectors are mapped to their closest discrete codewords, thereby reducing storage and memory bandwidth requirements while retaining expressive capacity. However, applying VQ directly to VLMs faces two fundamental challenges: (1) Modality-induced weight heterogeneity. In VLMs, image and text inputs induce divergent weight distributions, which a unified codebook fails to capture. (2) Error compensation mismatch from ignoring first-order gradients. In VLMs, first-order gradients significantly contribute to quantization error, yet conventional VQ methods neglect them, causing biased compensation and accuracy loss To this end, we propose **MSAVQ** (Multi-dimensional Sensitivity-Aware Vector Quantization), a framework that addresses these issues with two key components: (1) Sensitivity-driven structured mixed-precision quantization, a mixed-precision scheme that allocates bit-widths based on channel sensitivity, combining global and local saliency metrics for fine-grained and interpretable resource distribution. (2) Gradient-aware error compensation, a compensation method that explicitly incorporates first-order gradients to address their non-negligible role in VLM quantization errors, with efficient computation enabled by Kronecker and Block-LDL decompositions. We evaluate MSAVQ on representative VLMs, including LLaVA-onevision, InternVL2, and Qwen2-VL. In 2-bit settings, it consistently surpasses state-of-the-art PTQ methods, achieving up to **+4.9** higher accuracy (71.4% vs. 67.0% on InternVL2-26B). These results demonstrate that MSAVQ provides a simple and effective solution for ultra-low-bit quantization of multimodal foundation models, enabling practical deployment under strict resource budgets.

## 1 Introduction

Vision-Language Models (VLMs) are multimodal AI systems that integrate computer vision and natural language processing, taking both text and image/video inputs to generate text outputs, thereby enabling rich cross-modal reasoning and interaction (Bordes et al., 2024; Zhang et al., 2024a; Liu et al., 2023; Bai et al., 2023; Wang et al., 2024). While, these models typically contain billions of parameters, making training and inference computationally expensive and limiting their deployment in latency-sensitive or resource-constrained environments (Xue et al., 2025). For instance, Qwen2-VL-72B requires over 140GB of GPU memory during the prefill stage under FP16 inference, far exceeding the capacity of most edge devices. Reducing memory and bandwidth requirements

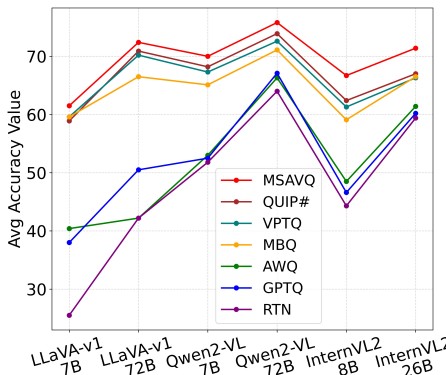

Figure 1: Comparison of average accuracy between MSAVQ and other quantization methods across different VLM models

while maintaining accuracy is therefore essential for practical deployment of large VLMs.

Post-training quantization (PTQ) avoids expensive retraining and substantially reduces storage and memory bandwidth, making it a key technique for compressing LLMs(Frantar et al., 2023; Li et al., 2025b; Xu et al., 2025). Currently, PTQ methods can be broadly categorized into two classes. Scalar quantization (SQ), which performs well at medium to high bit-widths ($\geq 4$ bits), assigns each weight an independent scaling factor and zero point, offering a lightweight representation (Frantar et al., 2023; Lin et al., 2024; Hu et al., 2024). However, as the bitwidth decreases to 3 bits or lower, the representational capacity of SQ becomes severely limited, resulting in sharp accuracy degradation. In contrast, vector quantization (VQ) maps high-dimensional weight vectors into a shared codebook, exploiting structural redundancy to achieve higher compression ratios (Gersho, 1979). This approach has been shown to substantially improve quantization performance under ultra-low-bitwidth settings (van Baalen et al., 2025; Liu et al., 2024a; Yue et al., 2025).

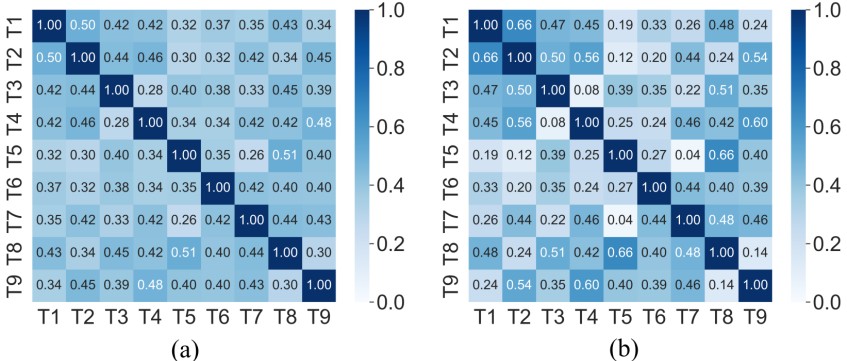

Figure 2: Similarity between tokens. (a) Text tokens similarity. (b) Image tokens similarity

However, directly applying vector quantization to VLMs leads to severe accuracy degradation due to two fundamental challenges: (1) Modality-induced weight heterogeneity. Within the same layer, VLM weights must simultaneously adapt to image and text tokens. As shown in Figure 2, these two types of tokens exhibit markedly different statistical characteristics, resulting in the heterogeneous weight distributions illustrated in Figure 3. Applying a unified codebook or fixed bit allocation across an entire layer fails to accommodate such structural heterogeneity, thereby amplifying quantization errors. (2) Error compensation mismatch from ignoring first-order gradients. As illustrated in Figure 4, first-order gradients in VLMs exhibit a highly concentrated distribution. However, prior VQ-based compensation methods (e.g., GPTVQ, VPTQ) typically disregard the first-order term and rely solely on second-order Taylor expansion, leading to inadequate error compensation and uneven error propagation across layers.

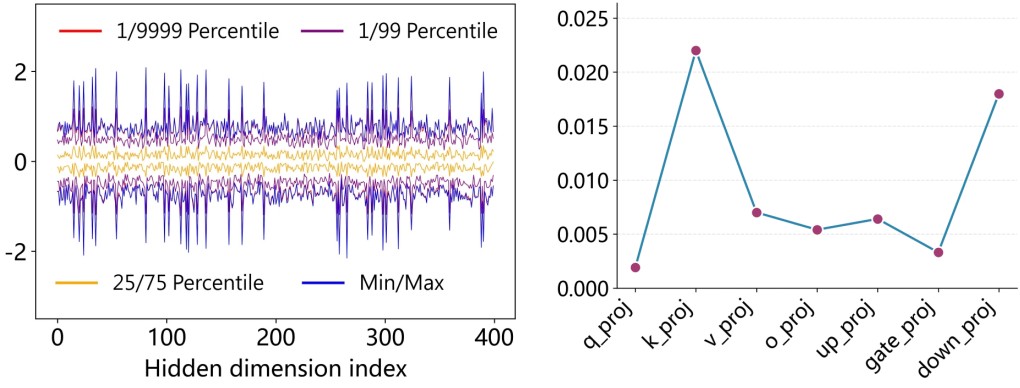

Figure 3: Weight distribution map of layer.1.down_proj in LLaVA-OneVision-7B

Figure 4: The gradient value at the 99% quantile of the gradient statistics of the 31st block of Qwen2-VL-72B

To address these limitations, we propose MSAVQ (Multi-dimensional Sensitivity-Aware Vector Quantization), a quantization framework consisting of two key components: (1) Sensitivity-driven structured mixed-precision quantization(SSMQ). We integrate both global and local sensitivity metrics to partition sub-blocks of weights and allocate optimal bit under a fixed bit budget. Highly

sensitive regions are assigned more bits, enabling fine-grained and interpretable resource allocation. (2) Gradient-aware error compensation(GAEC). For each layer, we perform a Taylor expansion of the global loss, where the quantization residual is used to approximate the first-order gradient matrix and the second-order Hessian is approximated via Kronecker factorization. Based on this formulation, we derive the theoretically optimal compensation rule and apply it iteratively to progressively reduce quantization errors.

These two components enable channel-level adaptive bit allocation and error compensation under resource-constrained conditions, while overcoming the limitation of conventional PTQ methods that neglect first-order terms. This significantly alleviates the accumulation of quantization errors in deep networks and their adverse impact during cross-modal propagation. Experimental results show that under 2-bit quantization, as observed in Figure 1, MSAVQ consistently outperforms existing approaches across multiple representative VLMs. For example, on the InternVL(Chen et al., 2024b) model, MSAVQ achieves more than a 4% improvement over QuIP# (Tseng et al., 2024b), and ablation studies further validate the independent contributions of each module.

The main contributions of this paper are summarized as follows:

- We identify two VLM-specific challenges for vector quantization: modality-induced weight heterogenei and error compensation mismatch from ignoring first-order gradients.

- We propose MSAVQ, which combines multi-dimensional sensitivity analysis, structured mixed-precision allocation, and gradient-aware error compensation to mitigate cross-layer and cross-modal quantization errors.

- We conduct extensive experiments on representative VLMs, showing that MSAVQ achieves superior accuracy under low bitwidth quantization while maintaining efficiency, outperforming existing state-of-the-art (SOTA) methods.

## 2 RELATED WORK

**Scalar Quantization (SQ)** maps parameters to uniformly spaced levels using a shared scaling factor and zero-point, implicitly assuming isotropy in the parameter space and uniform channel sensitivity. Current SQ schemes, when combined with auxiliary optimization techniques, have demonstrated strong performance at 4-bit precision and above. Representative approaches include GPTQ (Frantar et al., 2023) and GuidedQuant (Kim et al., 2025), which leverage Hessian-based error compensation to mitigate quantization loss. QuaRot (Ashkboos et al., 2024) and OstQuant (Hu et al., 2025) leverage rotation matrices to transform the parameter space, thereby improving the distribution of weights and activations across the quantization domain. MQuant (Yu et al., 2025) and MBQ (Li et al., 2025a) enhance multimodal PTQ by addressing modality disparities and outliers through structured techniques and gradient-based balancing, respectively, yielding improved accuracy and efficiency. While these methods are hardware-friendly and straightforward to implement, the quantization error grows sharply below 4-bit, limiting practicality at ultra-low bit allocation.

**Vector Quantization (VQ)** partitions weights into subvectors and approximates them using a codebook of limited prototypes. Compared with SQ, VQ offers stronger representational capacity and better accuracy retention under 3-bit and even lower precision. PCDVQ (Yue et al., 2025) decouples magnitude and direction in polar coordinates and uses distribution-aligned codebooks, achieving strong 2-bit performance. VPTQ (Liu et al., 2024a) employs channel-wise second-order optimization, efficient codebook initialization, and residual/outlier handling to achieve ultra-low-bit quantization, improving accuracy while reducing calibration time and boosting inference throughput. QuIP# (Tseng et al., 2024b) achieves state-of-the-art extreme compression by integrating structured transforms, lattice codebooks, and lightweight fine-tuning, enabling 3-bit models to outperform 4-bit baselines.

A comprehensive investigation of vector quantization (VQ) for VLMs remains absent, and a general-purpose framework has yet to be established. Two fundamental challenges underpin this gap. First, the modality-induced heterogeneity of weight distributions: visual and textual tokens exhibit markedly different statistical properties, resulting in mixed-distribution weights that cannot be effectively represented by a unified codebook. Second, the mismatch in error compensation arising from the neglect of first-order gradients: prevailing methods, such as GPTQ (Frantar et al., 2023) and YAQA (Tseng et al., 2025), operate under the assumption of near-zero gradients and consequently

restrict compensation to second-order, Hessian-based approximations, thereby leaving first-order contributions unaccounted for.

To tackle these gaps and challenges, we introduce MSAVQ. The method integrates multi-dimensional sensitivity analysis, structured mixed-precision allocation, and gradient-aware compensation, and is specifically designed to optimize ultra-low-bit quantization for VLMs.

## 3 PRELIMINARIES

**Vector Quantization in Post-Training Quantization.** In ultra-low-bit PTQ, VQ has attracted increasing attention due to its superior ability to model weight distributions and achieve higher compression ratios compared with scalar quantization. The core idea of VQ is to jointly encode correlated dimensions within small subspaces, approximating each weight sub-vector with a finite set of codewords. Formally, consider a weight matrix $W \in \mathbb{R}^{m \times n}$. Given a block size $v$ (with zero-padding applied if $v \nmid m\, n$), the matrix is reshaped into

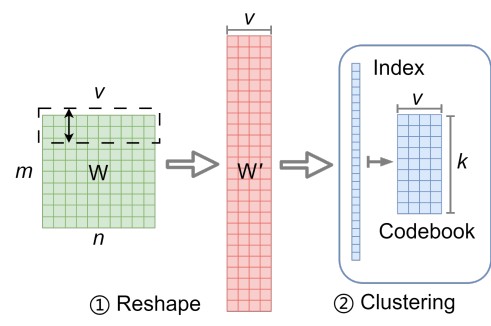

Figure 5: Vector Quantization.

$$W' \in \mathbb{R}^{M \times v}, \qquad M = \frac{m\, n}{v} \tag{1}$$

where the $i$-th row $W_i' \in \mathbb{R}^{1 \times v}$ corresponds to a weight vector block of length $v$.

A codebook $C \in \mathbb{R}^{K \times v}$ of size $K = 2^n$ is then constructed, where $n$ denotes the index bitwidth. Each vector block is quantized by selecting its nearest codeword from the codebook under Euclidean distance (noting that the Frobenius norm degenerates to $\ell_2$ distance in the vector case):

$$\mathrm{VQ}(W') = \left\{ j_i \mid j_i = \arg\min_{j \in \{1,\ldots,K\}} \left\| W_i' - C_j \right\|_2^2, \ \ i = 1, \ldots, M \right\} \tag{2}$$

Finally, the quantized weight matrix $\hat{W}$ is reconstructed by replacing each block with its assigned codeword and reshaping back to the original dimensions:

$$\hat{W} = \mathrm{reshape}(\hat{W}', m, n) \tag{3}$$

This formulation highlights how VQ leverages clustering in a shared codebook to exploit structural redundancy, thereby retaining stronger representational capacity under ultra-low-bit settings compared to scalar quantization.

When compressing models through quantization, pruning, or structural modification, it is essential to assess the global impact of such perturbations on model behavior. A straightforward approach is to directly measure changes in task-specific loss. Nevertheless, such an approach is often unreliable, as it depends heavily on the chosen evaluation dataset and cannot be easily decomposed across layers or parameters.

## 4 METHOD

The proposed MSAVQ (Multi-dimensional Sensitivity-Aware Vector Quantization) framework is built on the coordinated design of two key modules: channel-sensitivity-driven structured mixed-precision quantization and gradient-aware error compensation. To address multimodal inputs (image tokens and text tokens), MSAVQ first evaluates the sensirivity of input and output channels from both global sensitivity and local functional contribution, and fuses these metrics to form the basis for quantization resource allocation. The weight matrix is then reordered and partitioned into $2 \times 2$ structured sub-blocks, followed by closed-form bit allocation under a global bit budget, ensuring that highly sensitivity regions receive higher bitwidth. Finally, MSAVQ incorporates both first-order gradient and second-order hessian information to refine quantization results through fine-grained error compensation, achieving a balanced trade-off between model accuracy and compression efficiency. The detailed algorithmic procedures of the two modules are provided in Appendix A.7.

Figure 6: Overview of sensitivity-driven structured mixed-precision quantization(SSMQ)

## 4.1 SENSITIVITY-DRIVEN STRUCTURED MIXED-PRECISION QUANTIZATION(SSMQ)

The central challenge in mixed-precision quantization is how to allocate limited bit budgets so that critical parameters receive finer precision. To this end, we design a channel-sensitivity-driven structured quantization framework, as observed in Figure 6, which proceeds in three steps: channel sensitivity assessment (CSA), matrix reordering and structured block partitioning (MRSBP), and optimal bit allocation (OBA).

**Step 1: Channel Sensitivity Assessment (CSA)** We construct channel saliency by integrating global sensitivity and local sensitivity.

**Global sensitivity.** For a weight matrix $W \in \mathbb{R}^{m \times n}$ (with m output channels and n input channels), we adopt the Hessian of the KL divergence (equivalent to the Fisher Information Matrix) as a measure of global sensitivity(see Appendix A.4 for the derivation). It is approximated by a Kronecker factorization:

$$H \approx H_O \otimes H_I, H_O \in \mathbb{R}^{m \times m}, H_I \in \mathbb{R}^{n \times n} \tag{4}$$

where $H_O$ denotes the output-side Hessian capturing sensitivity along output channels, and $H_I$ denotes the input-side Hessian capturing sensitivity along input channels.

In practice, these components can be estimated from gradients at the sequence level:

$$H_I = \mathbb{E}\left[(\nabla_W \ell)^T (\nabla_W \ell)\right], \quad H_O = \mathbb{E}\left[(\nabla_W \ell)(\nabla_W \ell)^T\right] \tag{5}$$

where, $\nabla_W \ell$ denotes the gradient of the loss $\ell$ with respect to the weights $W$. The global sensitivity of input channel $i$ is defined as the $i$-th diagonal element of $H_I$, and the global sensitivity of output channel $j$ is defined as the $j$-th diagonal element of $H_O$. We denote these quantities as:

$$I_g^{(in)}[i] = H_I[i], I_g^{(out)}[j] = H_O[j] \tag{6}$$

**Local Sensitivity.** To measure the extent of local output influence across weight channels in practical scenarios, we compute the corresponding norm value of output activations as local sensitivity. The local sensitivity of input-output reflects the output energy generated by activation $x$ and weight $W$ across corresponding channels:

$$I_l^{(in)}[i] = \mathbb{E}\left[\|x \cdot W_{:,i}\|_2^2\right], I_l^{(out)}[j] = \mathbb{E}\left[\|x \cdot W_{j,:}\|_2^2\right] \tag{7}$$

**Combined Sensitivity.** The normalized global and local indicators are fused to obtain the final sensitivity scores:

$$I^{(in)}[i] = \log\left(\hat{I}_g^{(in)}[i] \cdot \hat{I}_l^{(in)}[i]\right), \quad I^{(out)}[j] = \log\left(\hat{I}_g^{(out)}[j] \cdot \hat{I}_l^{(out)}[j]\right) \tag{8}$$

where $\hat{\cdot}$ denotes min–max normalization. This fusion not only balances global sensitivity and local sensitivity, but also enables a relatively accurate assessment and quantification of quantization difficulty, thereby providing a reliable basis for bit allocation.

**Step 2: Matrix Reordering and Structured Block Partitioning (MRSBP)**

To cluster parameters with similar saliency, we sort input and output channels in descending order of $I^{(in)}$ and $I^{(out)}$. The reordered weight matrix $W'$ is then partitioned into $2 \times 2$ structured blocks. The saliency of each element $(i, j)$ is defined as the product of the corresponding input and output channel saliencies:

$$I_{i,j} = I^{(out)}[i] \cdot I^{(in)}[j] \tag{9}$$

Given partitioning cut points along the input/output dimensions, the matrix is divided into four sub-blocks $\{blk_t\}_{t=1}^4$. With a total bit budget B, the goal is to allocate bits $\{b_t\}$ to these sub-blocks to maximize the sensitivity–bit efficiency ratio:

$$\max_{\{b_t\}} \sum_{t=1}^4 \frac{S_t}{b_t}, \quad \text{s.t.} \sum_{t=1}^4 b_t = B, \ b_t > 0 \tag{10}$$

where $S_t = \sum_{(i,j)\in blk_t} I_{i,j}$ is the aggregated saliency of block $t$. This ensures that more sensitive regions receive finer quantization.

**Step 3: Optimal Bit Allocation (OBA)** Applying Lagrangian optimization yields the closed-form solution for the optimal bit allocation:

$$b_t = \frac{B \cdot \sqrt{S_t}}{\sum_{s=1}^4 \sqrt{S_s}} \tag{11}$$

This solution achieves the global optimum of Eq.10 in theory, ensuring the maximization of sensitivity return per bit of resource.The detailed derivation procedure is provided in Appendix A.2.

### 4.2 GRADIENT-AWARE ERROR COMPENSATION(GAEC)

**Step 1: Formulating the Joint First- and Second-Order Optimization Objective**

We frame the quantization problem by treating the residual between the original floating-point weights $W$ and their quantized counterparts $\hat{W}$ as the core optimization variable:

$$E \triangleq W - \hat{W}. \tag{12}$$

Optimizing $\hat{W}$ to minimize loss is mathematically equivalent to optimizing $E$, as the residual directly captures the discrepancy between the full-precision and quantized weights.

Our starting point is a joint first- and second-order optimization objective, which balances gradient alignment (first-order) and curvature-aware regularization (second-order):

$$\min_{\hat{W}} \text{vec}(E)^\top \nabla\mathcal{L} + \tfrac{1}{2}\text{vec}(E)^\top (H_O \otimes H_I)\text{vec}(E) \tag{13}$$

Here, $\nabla\mathcal{L}$ denotes the gradient of the loss with respect to $W$, and $H_O \otimes H_I$ represents the Kronecker product of Hessian blocks $H_O$ (output Hessian) and $H_I$ (input Hessian), encoding second-order curvature information. This objective leverages both local gradient direction and global curvature to guide quantization, striking a balance between alignment with loss reduction and stability. Theoretical justification of the Kronecker-factorized Hessian approximation and first-order gradient surrogate is provided in Appendix A.3.

To avoid the computational burden of explicit backpropagation for gradient estimation—especially critical in large-scale models—we approximate the loss gradient using the residual itself, scaled by a positive coefficient $\beta$:

$$\nabla\mathcal{L} \approx \beta E, \qquad \beta > 0. \tag{14}$$

This approximation is motivated by the observation that the residual $E$ often correlates with the gradient direction in practice, particularly when quantization errors dominate loss variations. Substituting this into the objective simplifies it to a strictly convex quadratic form in $E$:

$$\min_E \tfrac{1}{2}\langle E, (H_O \otimes H_I + 2\beta I)[E]\rangle, \tag{15}$$

where the added term $2\beta I$ acts as a diagonal regularizer, ensuring the overall operator $H_O \otimes H_I + 2\beta I$ is positive definite—critical for well-posedness and convergence (see Appendix A.3 for the error bound analysis).

To simplify optimization, we exploit the structure of the Hessian blocks via their (block) LDL decompositions:

$$H_O = (L_O + I)D_O(L_O + I)^\top, \qquad H_I = (L_I + I)D_I(L_I + I)^\top, \tag{16}$$

where $L_O, L_I$ are lower triangular matrices with zero diagonals, $D_O, D_I$ are diagonal matrices of positive entries, and $I$ is the identity matrix. We define $Z$ and $A^{(t)}$:

$$Z \triangleq (L_O + I)^\top E (L_I + I), \qquad A^{(t)} \triangleq (L_O + I)^{-T} E^{(t)} (L_I + I)^{-1}. \tag{17}$$

Then, Eq.15 can be decoupled element-wise into

$$\min_Z \sum_{i,j} \left[ \tfrac{1}{2} \lambda_{ij} Z_{ij}^2 \quad + \quad \beta A_{ij}^{(t)} Z_{ij} \right], \qquad \lambda_{ij} = D_O(i) D_I(j) > 0 \tag{18}$$

**Step 2: First-order correction term acquisition** Set the derivative of Eq.18 to zero, and the closed-form optimal solution is obtained:

$$Z_{ij}^\star = -\frac{\beta}{\lambda_{ij}} A_{ij}^{(t)}, \tag{19}$$

For improved numerical stability, we adopt a damped curvature $\tilde{\lambda}_{ij} = D_O(i) D_I(j) - 2\beta$ (ensuring positivity via $2\beta < \min_{i,j} D_O(i) D_I(j)$), which yields

$$Zij^\star = -\Gamma_{ij} A^{(t)}ij, \qquad \Gamma ij \triangleq \frac{\beta}{D_O(i) D_I(j) - 2\beta}. \tag{20}$$

Mapping back to the weight domain gives the **first-order correction term**

$$T \triangleq (L_O + I)^{-\top} \left[ \Gamma \circ A^{(t)} \right] (L_I + I)^{-1}, \tag{21}$$

where $\circ$ denotes the Hadamard product. When $2\beta \gg D_O(i) D_I(j)$, $\Gamma_{ij} \approx \frac{\beta}{D}_O(i) D_I(j)$, and $T \approx \beta P_O E^{(t)} P_I$ with $P_O = (L_O + I)^{-\top} D_O^{-1} (L_O + I)^{-\top}$, $P_I = (L_I + I)^{-1} D_I^{-1} (L_I + I)^{-1}$.

**Step 3: Fixed-Point Update Under Quantization Constraints (Projection)** We integrate second-order feedback and the above first-order correction into a single projection step. Let $E^{(t)} = W - \hat{W}^{(t)}$. Define the target

$$\eta \triangleq W + L_O^\top E^{(t)} L_I + L_O^\top E^{(t)} + E^{(t)} L_I - T, \tag{22}$$

and update by projection onto $\mathcal{Q}$:

$$\hat{W}^{(t+1)} \leftarrow \mathcal{Q}(\eta). \tag{23}$$

Repeat until convergence (recomputing $E^{(t)}$ each iteration). The damping condition $2\beta < \min_{i,j} D_O(i) D_I(j)$ guarantees well-posed element-wise scaling $\Gamma_{ij}$ and stabilizes the fixed-point iteration; as $\beta \to 0$, the method reduces to the purely second-order update without the first-order correction

## 5 EXPERIMENTS

### 5.1 EXPERIMENTAL SETTINGS

**Calibration set and evaluation benchmarks.** To collect the statistics required for quantization sensitivity analysis, we adopt the improved COCO image-caption dataset provided by ShareGPT4V (Chen et al., 2024a). A random subset of 128 image–text pairs is selected as the calibration set, which is used to compute Hessian information and channel importance scores. Model performance is evaluated with the LMMs-Eval benchmark suite (Zhang et al., 2024b) across a wide range of vision–language tasks, including: (1) Text recognition and understanding: OCRBench (Liu et al., 2024b) (scene text recognition), TextVQA(Singh et al., 2019) (text-centric visual question answering). (2) Visual perception: VizWiz (Gurari et al., 2018) (QA on everyday photos designed for visually impaired users), SEED-Bench (Li et al., 2024b) (a multimodal benchmark for generation and understanding). (3) Visual reasoning: ScienceQA (Lu et al., 2022) (science QA with multimodal inputs), MMMU Yue et al. (2024) (multi-discipline multimodal understanding and reasoning).

**Models and quantization settings.** We select three representative families of VLMs with both small and large versions to evaluate the generality of MSAVQ: LLaVA-onevision (Li et al., 2024a)(parameter sizes of 7B and 72B, with the VLM backbone based on Qwen2-7B/72B and the vision encoder SigLIP-400M (Zhai et al., 2023)). InternVL2 Chen et al. (2024b)(parameter sizes of 8B and 26B, with the VLM backbone InternLM2-8B/20B and the vision encoder InternViT-300M/6B). Qwen2-VL (Wang et al., 2024)(parameter sizes of 7B and 72B, with VLM backbone Qwen2-7B/72B and a vision encoder of 675M parameters). We evaluate 3-bit and 2-bit configurations for each model, comparing MSAVQ against strong PTQ baselines. Scalar quantization (SQ) baselines include RTN (uniform quantization), GPTQ (Frantar et al., 2023)(Hessian-guided), AWQ (Lin et al., 2024) (outlier-aware), and MBQ (Li et al., 2025a) (recent mixed-precision method). Vector quantization (VQ) baselines include VPTQ (Liu et al., 2024a) and QuIP# Tseng et al. (2024a), the latter being state-of-the-art for LLMs.

**Implementation details.** MSAVQ first performs row–column reordering of weight matrices, partitions them into four blocks, and applies vector quantization separately each block. The vector length is set to 4. K-means clustering is initialized with k-means++ and run for 100 iterations. All experiments are conducted on NVIDIA RTX A6000 GPUs.

## 5.2 MAIN RESULTS

Table 1: Under the 2 - bit and 3 - bit configurations of MSAVQ, a comparison is conducted between it and diverse quantization methods of VLMs.

| Bit | Method | LLaVA-onevision-7B | LLaVA-onevision-72B | Qwen2-VL-7B | Qwen2-VL-72B | InternVL2-8B | InternVL2-26B |
|---|---|---|---|---|---|---|---|
| FP16 | - | 66.9 | 74.3 | 73.1 | 78.1 | 71.7 | 74.6 |
| 3 | RTN | 47.9 | 72.1 | 65.4 | 75.0 | 69.0 | 73.3 |
| | GPTQ | 63.4 | 72.3 | 67.9 | 76.6 | 67.2 | 72.3 |
| | AWQ | 60.4 | 55.1 | 70.3 | 77.5 | 69.8 | 73.5 |
| | MBQ | 64.8 | 73.6 | 70.9 | 77.6 | 70.4 | 73.8 |
| | VPTQ | 65.3 | 73.6 | 71.3 | 77.5 | 70.6 | 73.9 |
| | MSAVQ | 65.8 | 74.0 | 71.9 | 77.8 | 71.1 | 74.2 |
| 2 | RTN | 25.5 | 42.2 | 51.8 | 64.0 | 44.3 | 59.4 |
| | GPTQ | 38 | 50.5 | 52.5 | 67.1 | 46.6 | 60.2 |
| | AWQ | 40.4 | 42.2 | 53.0 | 66.3 | 48.5 | 61.4 |
| | MBQ | 59.6 | 66.5 | 65.1 | 71.1 | 59.1 | 66.5 |
| | VPTQ | 59.6 | 70.2 | 67.3 | 72.6 | 61.3 | 66.3 |
| | QuIP# | 58.9 | 70.9 | 68.2 | 73.9 | 62.4 | 67.0 |
| | MSAVQ | 61.5 | 72.4 | 70.0 | 75.8 | 66.7 | 71.4 |

According to Table 1, we report the average accuracy across the six datasets introduced above. MSAVQ consistently achieves the best overall accuracy under both 3-bit and 2-bit quantization settings. In 3-bit quantization. MSAVQ outperforms the best existing baselines by 0.2–1.5 percentage points on average. For example, on Qwen2-VL-72B, MSAVQ achieves 77.8%, slightly higher than MBQ (77.6%) and VPTQ (77.5%). In 2-bit quantization. The advantage of MSAVQ is more pronounced, improving over the strongest baseline by 1.3–4.9 percentage points. On InternVL2-26B, MSAVQ reaches 71.4%, compared to QuIP# (67.0%) and MBQ (66.5%). MSAVQ also significantly narrows the gap between quantized and full-precision models. For instance, in the LLaVA-onevision-7B 2-bit case, the FP16 model achieves 66.9%, while MSAVQ reaches 61.5% (only $-5.4$). By contrast, the best baseline (VPTQ/MBQ) scores 59.6% ($-7.3$). This demonstrates MSAVQ's effectiveness in mitigating quantization-induced accuracy loss. In addition, end-to-end inference efficiency results are reported in Appendix A.5, which show consistent speedups across both prefilling and decoding stages. Detailed per-dataset results for each model are provided in Appendix A.6

## 5.3 ABLATION STUDIES

We conduct ablations on LLaVA-onevision-7B (Li et al., 2024a) and Qwen2-VL-7B (Wang et al., 2024), focusing on 2-bit and 3-bit scenarios across text recognition, visual perception, and visual reasoning tasks. We study the necessity and contributions of two core modules: CSMQ (channel-sensitivity-driven structured mixed-precision quantization). GSCM (gradient-enhanced second-order error compensation). Baselines include vanilla VQ (K-means), VPTQ, and GPTVQ.

**Joint effectiveness of CSMQ and GSCM.** As shown in Table 2, both modules are necessary and complementary. On LLaVA-onevision-7B (2-bit), without CSMQ/GSCM the average accuracy is

Table 2: Ablation experiments on CSMQ and GSMQ

| Model | Bit | CSMQ | GSCM | MMMU | SEED | OCRBench | VizWiz | ScienceQA | TextVQA | Average |
|-------|-----|------|------|------|------|----------|--------|-----------|---------|---------|
| LLaVA-onevision-7B | 2 | ✗ | ✗ | 33.1 | 51.3 | 50.1 | 51.0 | 73.1 | 61.0 | 53.3 |
| | | ✓ | ✗ | 38.9 | 65.1 | 52.3 | 55.8 | 82.1 | 67.9 | 60.4 |
| | | ✗ | ✓ | 35.2 | 56.9 | 51.4 | 53.9 | 76.9 | 66.3 | 56.8 |
| | | ✓ | ✓ | 40.1 | 66.3 | 54.5 | 56.1 | 83.1 | 68.9 | 61.5 |
| Qwen2-VL-7B | 3 | ✗ | ✗ | 41.4 | 63.2 | 68.3 | 61.4 | 81.1 | 71.4 | 64.5 |
| | | ✓ | ✗ | 47.2 | 68.3 | 73.9 | 66.3 | 83.1 | 77.9 | 69.5 |
| | | ✗ | ✓ | 45.9 | 68.9 | 69.6 | 65.6 | 82.2 | 76.9 | 68.2 |
| | | ✓ | ✓ | 49.3 | 70.5 | 78.1 | 68.1 | 84.3 | 80.9 | 71.9 |

only 53.3%. Adding CSMQ improves it to 60.4%, while GSCM alone yields 56.8%. Combining both achieves 61.5%, narrowing the FP16 gap to 5.4 and outperforming single-module gains by +1.2 and +4.7, respectively. On Qwen2-VL-7B (3-bit), the average score rises from 64.5% (no modules) to 71.9% (both modules), again showing strong synergy.

Table 3: Effectiveness of CSMQ

| Model | Bit | Method | MMMU | SEED | OCRBench | VizWiz | ScienceQA | TextVQA | Average |
|-------|-----|--------|------|------|----------|--------|-----------|---------|---------|
| LLaVA-onevision-7B | 2 | Kmeans | 33.1 | 51.3 | 50.1 | 51.0 | 73.1 | 61.0 | 53.3 |
| | | VPTQ | 38.7 | 64.6 | 51.1 | 55.3 | 80.4 | 67.3 | 59.6 |
| | | OURS(CSMQ) | 38.9 | 65.1 | 52.3 | 55.8 | 82.1 | 67.9 | 60.4 |

**Effectiveness of CSMQ.** Table 3 compares CSMQ with existing VQ methods. On LLaVA-onevision-7B (2-bit), vanilla VQ achieves 53.3%, VPTQ scores 59.6%, while CSMQ reaches 60.4%. Notably, in ScienceQA, accuracy improves from 73.1% (K-means) to 82.1%, and in TextVQA from 61.0% to 67.9%, validating CSMQ's advantage in dynamically allocating precision to channels under hybrid-distribution weights in VLMs.

Table 4: Effectiveness of GSCM

| Model | Bit | Method | MMMU | SEED | OCRBench | VizWiz | ScienceQA | TextVQA | Average |
|-------|-----|--------|------|------|----------|--------|-----------|---------|---------|
| Qwen2-VL-7B | 2 | GPTVQ | 43.9 | 65.2 | 67.1 | 63.2 | 80.1 | 74.5 | 65.7 |
| | | VPTQ | 44.9 | 68.1 | 67.2 | 65.6 | 81.1 | 76.9 | 67.3 |
| | | OURS(GSCM) | 45.9 | 68.9 | 69.6 | 65.6 | 82.2 | 76.9 | 68.2 |

**Effectiveness of GSCM.** From Table 4, GSCM improves over GPTVQ and VPTQ by explicitly incorporating first-order gradient terms into second-order error compensation. Traditional second-order approaches such as GPTVQ (65.7%) underestimate small gradient regions (0–0.001), leading to insufficient compensation. GSCM alleviates this by leveraging gradient residuals, achieving 68.2% on Qwen2-VL-7B (3-bit), surpassing GPTVQ (65.7%) and VPTQ (67.3%). Gains are especially evident on gradient-sensitive tasks: OCRBench improves from 67.1% to 69.6%, and MMMU from 43.9% to 45.9%, demonstrating reduced error accumulation across layers and modalities.

## 6 CONCLUSION

This work addresses the unique challenges of applying vector quantization to vision-language models (VLMs), where modality-induced weight heterogeneity and the non-negligible role of first-order gradients lead to severe performance degradation under low-bit settings. We propose **MSAVQ**, a multi-dimensional saliency-aware vector quantization framework that integrates (1) modality-induced weight heterogeneit, and (2) gradient-aware error compensation. By jointly leveraging global-local sensitivity measures and efficient Kronecker/Block-LDL decomposition, MSAVQ achieves fine-grained bit allocation and accurate error correction. Extensive experiments across diverse VLM families (LLaVA-onevision, InternVL2, Qwen2-VL) and model scales (7B–72B) demonstrate that MSAVQ consistently outperforms existing SQ and VQ baselines, especially in the extreme 2-bit regime. Notably, MSAVQ significantly reduces the quantization–FP16 gap, highlighting its effectiveness in mitigating cross-modal error accumulation. Ablation studies further confirm the complementary contributions of sensitivity-driven allocation and gradient-aware compensation. Our findings suggest that well-founded quantization strategies are crucial for enabling efficient deployment of large-scale multimodal models. Beyond the immediate improvements in VLM quantization, the proposed MSAVQ framework offers a general perspective on integrating structural sensitivity analysis and gradient-informed optimization, which may inspire future research on compressing and accelerating multimodal foundation models. In the future, we plan to extend MSAVQ to more complex multimodal scenarios, such as video-language understanding and multi-task joint modeling, to further explore its generalization potential.

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

# A APPENDIX

## A.1 USE OF LLMS

In preparing this manuscript, large language model (LLM) tools were applied only as auxiliary aids for improving linguistic expression—such as enhancing clarity, polishing phrasing, and refining overall readability.

The originality and scientific substance of the work rest entirely on the efforts of the research team. Specifically, the development of the research framework, the design and implementation of algorithms, the setup and execution of experiments, the handling and analysis of data, and the verification of findings were all carried out independently by the authors.

No part of the conceptualization of research content, the creation of technical approaches, the execution of experimental work, or the derivation of conclusions involved the use of LLM tools. The authors fully guarantee the integrity, authenticity, and originality of this study, in line with academic ethical standards.

## A.2 CLOSED-FORM SOLUTION FOR OPTIMAL BIT ALLOCATION

The optimal bit allocation $\{b_t\}_{t=1}^4$ that minimizes the objective

$$\min_{\{b_t\}} \sum_{t=1}^4 \frac{S_t}{b_t}, \quad \text{s.t.} \sum_{t=1}^4 b_t = B, \; b_t > 0, \tag{24}$$

where $S_t = \sum_{(i,j)\in\text{blk}_t} I_{i,j}$ is the total saliency of block $t$, is given by the closed-form expression:

$$b_t = \frac{B \cdot \sqrt{S_t}}{\sum_{s=1}^4 \sqrt{S_s}}, \quad t = 1, \ldots, 4. \tag{25}$$

*Proof.* Define the Lagrangian:

$$\mathcal{L}(b_1, \ldots, b_4, \lambda) = \sum_{t=1}^4 \frac{S_t}{b_t} + \lambda \Big( B - \sum_{t=1}^4 b_t \Big). \tag{26}$$

Taking the derivative with respect to $b_t$ and setting it to zero gives:

$$\frac{\partial \mathcal{L}}{\partial b_t} = -\frac{S_t}{b_t^2} - \lambda = 0 \; \Rightarrow \; \frac{S_t}{b_t^2} = -\lambda. \tag{27}$$

Hence,

$$\frac{S_1}{b_1^2} = \cdots = \frac{S_4}{b_4^2} = c, \tag{28}$$

for some constant $c$, leading to

$$b_t = \sqrt{\frac{S_t}{c}}. \tag{29}$$

Applying the constraint $\sum_{t=1}^4 b_t = B$, we obtain

$$c = \left( \frac{\sum_{t=1}^4 \sqrt{S_t}}{B} \right)^2. \tag{30}$$

Substituting $c$ yields the closed-form solution

$$b_t = \frac{B \cdot \sqrt{S_t}}{\sum_{s=1}^4 \sqrt{S_s}}. \tag{31}$$

This ensures that more bits are allocated to blocks with larger saliency, achieving globally optimal efficiency.

### A.3 ERROR BOUND UNDER KRONECKER HESSIAN AND FIRST-ORDER GRADIENT APPROXIMATION

Assume the layerwise Hessian admits a Kronecker-factored approximation $H \approx H_O \otimes H_I$ with block–LDL factors $H_O = (L_O + I)D_O(L_O + I)^\top$, $H_I = (L_I + I)D_I(L_I + I)^\top$, and the gradient is approximated by $g \approx \beta E$ with error $\|\nabla\mathcal{L} - \beta E\| \leq \gamma\|E\|$. Then the (expected) quantization error satisfies

$$\mathcal{E} \leq \frac{\operatorname{tr}(D_O)\operatorname{tr}(D_I) + \gamma}{\beta + \lambda_{\min}(H_O \otimes H_I)}. \tag{32}$$

*Proof.* We analyze the standard quadratic surrogate for the layerwise loss around the full-precision weights with both second- and first-order terms retained:

$$\mathcal{J}(E) = \tfrac{1}{2}\langle E, (H_O \otimes H_I)E\rangle + \langle g, E\rangle, \tag{33}$$

where $E = W - \hat{W}$ is the quantization error. To avoid explicit backpropagation of gradients, the gradient term is approximated by the quantization residual with a scaling factor:

$$g \approx \beta E + \delta, \qquad \|\delta\| \leq \gamma\|E\|, \tag{34}$$

where $\beta$ balances the scale and $\delta$ denotes the bounded approximation error. This approximation can be motivated by a first-order Taylor expansion, in which the quantization residual serves as a surrogate for the gradient while $\delta$ captures the residual perturbation.

Substituting $g = \beta E + \delta$ gives

$$\mathcal{J}(E) = \tfrac{1}{2}\langle E, (H_O \otimes H_I)E\rangle + \beta\|E\|^2 + \langle\delta, E\rangle. \tag{35}$$

By Cauchy–Schwarz and the perturbation bound on $\delta$ we obtain

$$\mathcal{J}(E) \geq \tfrac{1}{2}\lambda_{\min}(H_O \otimes H_I)\|E\|^2 + \beta\|E\|^2 - \|\delta\|\|E\| \geq \left(\tfrac{1}{2}\lambda_{\min}(H_O \otimes H_I) + \beta - \gamma\right)\|E\|^2. \tag{36}$$

Equivalently, if we absorb the conventional $\tfrac{1}{2}$ into the Hessian surrogate (i.e., work with $A = H_O \otimes H_I$ in the quadratic model), we have the cleaner strong-convexity lower bound

$$\mathcal{J}(E) \geq \left(\lambda_{\min}(H_O \otimes H_I) + \beta\right)\|E\|^2 - \gamma\|E\|^2 = \left(\beta + \lambda_{\min}(H_O \otimes H_I) - \gamma\right)\|E\|^2. \tag{37}$$

Next, we upper-bound the contribution of the stochastic rounding term in the quadratic form. If $H_O = (L_O + I)D_O(L_O + I)^\top$ and $H_I = (L_I + I)D_I(L_I + I)^\top$ are the block–LDL factors, then the expected quadratic error satisfies

$$\mathbb{E}\big[\langle E, (H_O \otimes H_I)E\rangle\big] \leq \operatorname{tr}(D_O)\operatorname{tr}(D_I)\sigma^2, \tag{38}$$

for stochastic rounding noise with variance proxy $\sigma^2$. This establishes a trace-based control of the second-order contribution under Kronecker-factored Hessians.

Putting the pieces together and dropping the negligible variance scaling (or normalizing so $\sigma^2 = 1$), we obtain an expected upper bound on the numerator of $\mathcal{J}(E)$:

$$\mathbb{E}[\mathcal{J}(E)] \leq \tfrac{1}{2}\operatorname{tr}(D_O)\operatorname{tr}(D_I) + \beta\|E\|^2 + \gamma\|E\|^2. \tag{39}$$

Combining with the strong-convexity lower bound yields, up to the same scaling convention as above,

$$\left(\beta + \lambda_{\min}(H_O \otimes H_I)\right)\|E\|^2 \lesssim \operatorname{tr}(D_O)\operatorname{tr}(D_I) + \gamma. \tag{40}$$

Defining the (expected) quantization error metric $\mathcal{E} = \|E\|^2$ we obtain

$$\mathcal{E} \leq \frac{\operatorname{tr}(D_O)\operatorname{tr}(D_I) + \gamma}{\beta + \lambda_{\min}(H_O \otimes H_I)}, \tag{41}$$

which is the claimed bound. The role of the Hessian Kronecker approximation accuracy (denoted as $\epsilon$ in the main text) is to mildly inflate the term $\operatorname{tr}(D_O)\operatorname{tr}(D_I)$, and can be absorbed into the numerator.

## A.4 GLOBAL SENSITIVITY VIA KL DIVERGENCE

In model compression, directly evaluating task loss changes is often unreliable, as it depends on specific datasets and cannot be decomposed across layers. Instead, we adopt the Kullback–Leibler (KL) divergence between the outputs of the full-precision model and the quantized model as a principled measure of quantization sensitivity.

Given a full-precision model $M(W, X)$ with parameters $W$ and its quantized counterpart $M(\hat{W}, X)$, the global KL loss is defined as:

$$\mathcal{L}_{\mathrm{KL}}(\hat{W}) = \mathbb{E}_{X \sim \mathcal{D}} D_{\mathrm{KL}}(M(W, X) \| M(\hat{W}, X)) \tag{42}$$

where $X \sim \mathcal{D}$ denotes inputs drawn from the data distribution $D$. For small perturbations around $W$, a second-order Taylor expansion yields:

$$\mathcal{L}_{\mathrm{KL}}(\hat{W}) \approx \frac{1}{2}(\hat{W} - W)^{\top} H_{\mathrm{global}}(\hat{W} - W) \tag{43}$$

where $H_{global}$ is the Hessian of the KL divergence, equivalent to the Fisher Information Matrix:

$$H_{\mathrm{global}} = \mathbb{E}\left[\nabla_W \ell \, \nabla_W \ell^{\top}\right] \tag{44}$$

Thus, the global Hessian provides a second-order sensitivity metric for weight perturbations, forming the theoretical basis for saliency analysis and bit allocation in our quantization framework.

## A.5 ADDITIONAL RESULTS

Table 5: The end-to-end speed up of LLaVA-onevision-7B on RTX4090 with fused GPU kernels. Experimental results show that our method achieves approximately 9% (W3) and 15% (W2) acceleration over FP16 on the Vision Transformer (ViT), delivers an average improvement of about 7–13% in the VLM Prefill stage, and reaches up to 37% acceleration in the Decoder stage, demonstrating the inference efficiency of our approach.

| Model | Stage | FP16 (ms) | W3 (ms) | W2 (ms) |
|---|---|---|---|---|
| ViT | Prefill (729 tokens) | 11.5 | 10.5 | 9.7 |
| VLM | Prefill (512 tokens) | 68.1 | 61.3 | 59.8 |
| | Prefill (1024 tokens) | 108.6 | 100.5 | 94.7 |
| | Decode | 29.3 | 18.4 | 25.6 |

## A.6 ALL RESULTS

Table 6: Results of LLaVA-onevision-7B.

| Bit | Method | MMMU | SEED | OCRBench | VizWiz | ScienceQA | TextVQA | Average (↑) |
|---|---|---|---|---|---|---|---|---|
| FP16 | - | 46.0 | 71.1 | 62.2 | 60.4 | 85.4 | 76.1 | 66.9 |
| 3 | RTN | 34.7 | 10.4 | 35.9 | 59.2 | 86.2 | 60.9 | 47.9 |
| | GPTQ | 41.9 | 68.7 | 55.7 | 56.4 | 86.4 | 71.3 | 63.4 |
| | AWQ | 36.6 | 51.5 | 59.3 | 58.5 | 83.2 | 73.0 | 60.4 |
| | MBQ | 42.0 | 66.4 | 61.1 | 60.7 | 85.0 | 73.3 | 64.8 |
| | VPTQ | 43.3 | 69.1 | 61.4 | 60.2 | 84.7 | 73.1 | 65.3 |
| | MSAVQ | 44.0 | 69.9 | 61.5 | 60.3 | 85.1 | 73.9 | 65.8 |
| 2 | RTN | 13.9 | 0.0 | 10.3 | 36.8 | 61.3 | 30.4 | 25.5 |
| | GPTQ | 30.2 | 8.5 | 29.7 | 50.1 | 70.3 | 39.4 | 38.0 |
| | AWQ | 30.9 | 9.8 | 35.3 | 50.3 | 71.9 | 43.9 | 40.4 |
| | MBQ | 37.6 | 63.5 | 52.0 | 56.1 | 81.2 | 67.5 | 59.6 |
| | VPTQ | 38.7 | 64.6 | 51.1 | 55.3 | 80.4 | 67.3 | 59.6 |
| | QuIP# | 37.7 | 62.3 | 51.0 | 55.2 | 80.3 | 67.1 | 58.9 |
| | MSAVQ | 40.1 | 66.3 | 54.5 | 56.1 | 83.1 | 68.9 | 61.5 |

Table 7: Results of LLaVA-onevision-72B.

| Bit | Method | MMMU | SEED | OCRBench | VizWiz | ScienceQA | TextVQA | Average (↑) |
|---|---|---|---|---|---|---|---|---|
| FP16 | - | 56.1 | 78.1 | 73.2 | 69.2 | 90.0 | 79.3 | 74.3 |
| 3 | RTN | 53.9 | 77.4 | 68.2 | 66.1 | 89.5 | 77.4 | 72.1 |
|  | GPTQ | 52.7 | 76.0 | 69.7 | 68.3 | 89.3 | 77.9 | 72.3 |
|  | AWQ | 33.4 | 71.2 | 48.7 | 49.3 | 69.2 | 58.8 | 55.1 |
|  | MBQ | 54.4 | 77.6 | 71.6 | 69.0 | 90.3 | 78.5 | 73.6 |
|  | VPTQ | 54.5 | 77.8 | 71.9 | 69.1 | 90.0 | 78.4 | 73.6 |
|  | MSAVQ | 55.6 | 77.9 | 72.5 | 69.0 | 90.1 | 79.0 | 74.0 |
| 2 | RTN | 34.5 | 18.5 | 34.5 | 50.1 | 71.1 | 44.4 | 42.2 |
|  | GPTQ | 47.2 | 30.1 | 40.3 | 58.4 | 74.0 | 52.9 | 50.5 |
|  | AWQ | 33.0 | 17.9 | 31.2 | 54.9 | 69.2 | 47.1 | 42.2 |
|  | MBQ | 48.1 | 70.4 | 67.1 | 60.2 | 83.8 | 69.1 | 66.5 |
|  | VPTQ | 51.3 | 74.6 | 69.0 | 66.3 | 86.8 | 72.9 | 70.2 |
|  | QuIP# | 52.5 | 75.3 | 69.9 | 66.5 | 86.8 | 74.6 | 70.9 |
|  | MSAVQ | 53.4 | 75.8 | 71.7 | 68.1 | 87.9 | 77.4 | 72.4 |

Table 8: Results of InternVL2-8B.

| Bit | Method | MMMU | SEED | OCRBench | VizWiz | ScienceQA | TextVQA | Average (↑) |
|---|---|---|---|---|---|---|---|---|
| FP16 | - | 48.0 | 71.6 | 76.5 | 61.1 | 96.2 | 77.0 | 71.7 |
| 3 | RTN | 43.7 | 70.3 | 74.0 | 56.0 | 95.6 | 74.6 | 69.0 |
|  | GPTQ | 41.7 | 68.9 | 70.2 | 59.9 | 89.5 | 73.1 | 67.2 |
|  | AWQ | 44.8 | 70.4 | 74.7 | 58.9 | 95.5 | 74.2 | 69.8 |
|  | MBQ | 46.9 | 70.8 | 75.1 | 58.7 | 95.6 | 75.1 | 70.4 |
|  | VPTQ | 47.1 | 70.9 | 75.4 | 59.1 | 95.5 | 75.8 | 70.6 |
|  | MSAVQ | 47.6 | 71.3 | 75.9 | 59.5 | 95.6 | 76.5 | 71.1 |
| 2 | RTN | 33.5 | 10.2 | 34.1 | 50.9 | 72.2 | 65.1 | 44.3 |
|  | GPTQ | 30.4 | 18.9 | 37.9 | 48.1 | 77.9 | 66.3 | 46.6 |
|  | AWQ | 34.5 | 20.7 | 38.2 | 53.2 | 75.8 | 68.8 | 48.5 |
|  | MBQ | 40.3 | 65.8 | 50.4 | 53.3 | 77.3 | 67.3 | 59.1 |
|  | VPTQ | 44.9 | 64.9 | 57.3 | 55.2 | 77.1 | 68.1 | 61.3 |
|  | QuIP# | 45.3 | 67.3 | 61.2 | 54.1 | 78.2 | 68.4 | 62.4 |
|  | MSAVQ | 46.2 | 69.3 | 68.4 | 57.6 | 86.4 | 72.3 | 66.7 |

Table 9: Results of InternVL2-26B.

| Bit | Method | MMMU | SEED | OCRBench | VizWiz | ScienceQA | TextVQA | Average (↑) |
|---|---|---|---|---|---|---|---|---|
| FP16 | - | 47.1 | 76.8 | 77.9 | 66.2 | 97.5 | 82.1 | 74.6 |
| 3 | RTN | 46.6 | 75.7 | 75.9 | 64.7 | 96.4 | 80.6 | 73.3 |
|  | GPTQ | 44.8 | 75.8 | 76.0 | 60.9 | 96.3 | 80.1 | 72.3 |
|  | AWQ | 46.4 | 76.2 | 76.4 | 64.5 | 96.7 | 81.0 | 73.5 |
|  | MBQ | 47.1 | 76.3 | 76.5 | 64.5 | 97.3 | 81.1 | 73.8 |
|  | VPTQ | 47.3 | 76.0 | 76.9 | 65.2 | 97.1 | 81.0 | 73.9 |
|  | MSAVQ | 47.1 | 76.4 | 77.3 | 65.0 | 97.3 | 81.8 | 74.2 |
| 2 | RTN | 36.2 | 55.6 | 63.5 | 56.1 | 73.4 | 71.4 | 59.4 |
|  | GPTQ | 37.4 | 58.2 | 64.3 | 55.1 | 72.4 | 73.8 | 60.2 |
|  | AWQ | 38.4 | 55.4 | 64.9 | 57.4 | 74.9 | 77.4 | 61.4 |
|  | MBQ | 43.2 | 65.3 | 70.2 | 58.4 | 83.9 | 78.1 | 66.5 |
|  | VPTQ | 42.4 | 67.4 | 72.3 | 55.1 | 80.3 | 80.1 | 66.3 |
|  | QuIP# | 44.2 | 69.3 | 71.2 | 56.1 | 80.3 | 81.1 | 67.0 |
|  | MSAVQ | 46.1 | 73.9 | 73.2 | 60.3 | 93.6 | 81.1 | 71.4 |

Table 10: Results of Qwen2-VL-7B.

| Bit | Method | MMMU | SEED | OCRBench | VizWiz | ScienceQA | TextVQA | Average (↑) |
|---|---|---|---|---|---|---|---|---|
| FP16 | - | 50.6 | 71.9 | 80.7 | 68.3 | 85.1 | 82.0 | 73.1 |
| 3 | RTN | 44.9 | 69.8 | 60.0 | 65.2 | 81.5 | 71.2 | 65.4 |
| | GPTQ | 43.1 | 68.9 | 74.8 | 64.3 | 79.7 | 76.7 | 67.9 |
| | AWQ | 44.7 | 70.4 | 76.9 | 68.0 | 82.5 | 79.5 | 70.3 |
| | MBQ | 47.9 | 70.2 | 76.8 | 67.7 | 82.8 | 79.9 | 70.9 |
| | VPTQ | 48.3 | 70.5 | 77.6 | 67.9 | 83.4 | 79.9 | 71.3 |
| | MSAVQ | 49.3 | 70.5 | 78.1 | 68.1 | 84.3 | 80.9 | 71.9 |
| 2 | RTN | 36.5 | 55.9 | 50.4 | 45.9 | 71.8 | 50.3 | 51.8 |
| | GPTQ | 37.3 | 57.2 | 50.7 | 44.8 | 70.9 | 54.3 | 52.5 |
| | AWQ | 38.7 | 58.1 | 51.0 | 44.3 | 70.8 | 55.2 | 53.0 |
| | MBQ | 43.9 | 67.3 | 59.8 | 66.4 | 81.2 | 72.2 | 65.1 |
| | VPTQ | 44.9 | 68.1 | 67.2 | 65.6 | 81.1 | 76.9 | 67.3 |
| | QuIP# | 45.6 | 69.0 | 66.9 | 66.4 | 83.1 | 78.4 | 68.2 |
| | MSAVQ | 46.8 | 68.9 | 74.9 | 67.1 | 83.4 | 79.0 | 70.0 |

Table 11: Results of Qwen2-VL-72B.

| Bit | Method | MMMU | SEED | OCRBench | VizWiz | ScienceQA | TextVQA | Average (↑) |
|---|---|---|---|---|---|---|---|---|
| FP16 | - | 61.1 | 77.6 | 79.9 | 76.0 | 91.6 | 82.5 | 78.1 |
| 3 | RTN | 57.7 | 77.5 | 70.4 | 74.8 | 89.7 | 79.7 | 75.0 |
| | GPTQ | 57.3 | 77.2 | 78.5 | 73.6 | 91.5 | 81.6 | 76.6 |
| | AWQ | 59.6 | 77.6 | 79.6 | 75.4 | 90.4 | 82.4 | 77.5 |
| | MBQ | 59.6 | 77.7 | 79.4 | 75.6 | 90.5 | 82.5 | 77.6 |
| | VPTQ | 59.4 | 77.6 | 79.0 | 75.8 | 90.9 | 82.1 | 77.5 |
| | MSAVQ | 60.6 | 77.7 | 79.3 | 75.8 | 91.4 | 82.2 | 77.8 |
| 2 | RTN | 42.1 | 66.9 | 61.3 | 62.3 | 80.2 | 71.3 | 64.0 |
| | GPTQ | 44.2 | 68.1 | 66.3 | 65.4 | 82.5 | 75.8 | 67.1 |
| | AWQ | 44.5 | 67.3 | 66.9 | 64.3 | 80.4 | 74.5 | 66.3 |
| | MBQ | 48.9 | 71.4 | 74.9 | 69.8 | 82.9 | 78.4 | 71.1 |
| | VPTQ | 53.2 | 73.4 | 76.4 | 69.1 | 83.4 | 79.8 | 72.6 |
| | QuIP# | 55.8 | 73.9 | 76.1 | 72.3 | 85.8 | 79.5 | 73.9 |
| | MSAVQ | 58.8 | 75.9 | 78.0 | 73.1 | 87.9 | 81.3 | 75.8 |

## A.7 ALGORITHM

---

**Algorithm 1** SSMQ: Sensitivity-driven Structured Mixed-precision Quantization

---

**Input:** Weight matrix $W \in \mathbb{R}^{m \times n}$, total bit budget $B$
**Output:** Quantized weights $\hat{W}$

**CSA: Channel Sensitivity Assessment**

1: Compute Hessian factors $H_I, H_O$ via Kronecker approximation      ▷ Eq. 5
2: **for** each input/output channel **do**
3:      Compute global sensitivity ($H_I, H_O$ diag), local sensitivity (activation norm)    ▷ Eq. 6 7
4:      Fuse normalized scores: $I^{(in/out)} = \log(\hat{I}_g^{(in/out)} \cdot \hat{I}_l^{(in/out)})$      ▷ Eq. 8
5: **end for**

**MRSBP: Reordering & Partitioning**

6: Sort channels by $I^{(in)}, I^{(out)}$, define saliency $I_{i,j} = I^{(out)}[i] \cdot I^{(in)}[j]$      ▷ Eq. 9
7: Partition $W$ into 4 blocks and compute block saliency $S_t$      ▷ Eq. 10

**OBA: Optimal Bit Allocation**

8: **for** each block $t$ **do**
9:      $b_t = B \cdot \frac{\sqrt{S_t}}{\sum_{s=1}^{4} \sqrt{S_s}}$, then quantize block with $b_t$ bits      ▷ Eq. 11
10: **end for**
11: **return** $\hat{W}$

---

---

**Algorithm 2** GAEC: Gradient-aware Error Compensation

---

**Input:** Original weights $W$, Hessian blocks $H_O, H_I$, quantizer $\mathcal{Q}$, scaling factor $\beta$, tolerance $\varepsilon$, max iterations $T$
**Output:** Optimized quantized weights $\hat{W}$

**Initialization**

1: $(L_O, D_O) = \text{BlockLDL}(H_O), \quad (L_I, D_I) = \text{BlockLDL}(H_I)$
2: $\hat{W} = \mathcal{Q}(W), \quad t = 0$

**Iterative Compensation**

3: **while** $t < T$ and not converged **do**
4:      $E = W - \hat{W}$      ▷ Eq. 12
5:      $A = (L_O + I)^{-\top} E (L_I + I)^{-1}$      ▷ Eq. 17
6:      $\Gamma_{ij} = \dfrac{\beta}{D_O(i)D_I(j) - 2\beta}$      ▷ Eq. 20
7:      $T = (L_O + I)^{-\top} (\Gamma \circ A)(L_I + I)^{-1}$      ▷ Eq. 21
8:      $\eta \le W + L_O^\top E L_I + L_O^\top E + E L_I - T$      ▷ Eq. 22
9:      $\hat{W}_{new} = \mathcal{Q}(\eta)$      ▷ Eq. 23
10:      $\Delta = \|\hat{W}_{new} - \hat{W}\|_F / \max(1, \|\hat{W}\|_F)$
11:      **if** $\Delta < \varepsilon$ **then**
12:          break
13:      **end if**
14:      $\hat{W} = \hat{W}_{new}, \quad t = t + 1$
15: **end while**
16: **return** $\hat{W}$

---

