# OpenReview forum: "MSAVQ: Multi-dimensional Sensitivity-Aware Vector Quantization for Ultra-Low-Bit Vision-Language Models"
_ICLR.cc/2026/Conference — ICLR 2026 Conference Desk Rejected Submission_

### Official Review · Reviewer_iNQF · 2025-11-01

**Soundness:** 2
**Presentation:** 3
**Contribution:** 2
**Rating:** 6
**Confidence:** 3

**Summary:**

MSAVQ proposes a multi-dimensional sensitivity-aware vector quantization framework for Vision-Language Models (VLMs). It integrates two key modules—channel-sensitivity-driven structured mixed-precision quantization (SSMQ) and gradient-aware error compensation (GAEC)—to significantly improve quantization accuracy under ultra-low bit settings (2–3 bits). The method consistently outperforms existing state-of-the-art PTQ approaches across multiple representative VLMs, including LLaVA, InternVL2, and Qwen2-VL.

**Strengths:**

MSAVQ has a well-founded optimization theory, jointly modeling first- and second-order error terms and deriving a closed-form update rule that theoretically guarantees convergence and numerical stability. It enables efficient implementation, requiring only a small calibration set and simple K-means clustering without any retraining, making it practical and easily reproducible.

**Weaknesses:**

The paper lacks hardware-level validation, failing to evaluate the practical deployability of SSMQ and GAEC on real hardware, while the proposed channel-wise adaptive bit allocation may introduce additional storage or computation overhead that remains unaddressed. Meanwhile, its baseline selection is limited—though the method surpasses traditional PTQ approaches such as QuIP, it lacks comparison with more recent quantization methods, which weakens the experimental rigor.

**Questions:**

1. Have you conducted validation on real hardware to assess the practical deployability? If so, please provide details on storage/computation overhead introduced by the channel-wise adaptive bit allocation; if not, please explain the reasons and supplement relevant evaluations.
2. It is necessary to provide performance comparisons with more recent quantization methods. Please supplement these comparative experiments and analyze the performance differences between the proposed method and these methods.

---

### Official Review · Reviewer_upkj · 2025-11-01

**Soundness:** 2
**Presentation:** 2
**Contribution:** 2
**Rating:** 2
**Confidence:** 3

**Summary:**

The paper proposes MSAVQ, a post-training vector-quantization framework for VLMs that (i) computes multi-dimensional channel sensitivity, (ii) performs structured mixed-precision bit allocation via a closed-form rule, and (iii) applies a gradient-aware error compensation step using a Kronecker-factored Hessian with a first-order surrogate of the gradient and a damped fixed-point projection under quantization constraints.

**Strengths:**

- Clear formulation of the VQ/PTQ setup with straightforward notation.

- The closed-form bit-allocation subproblem is convex and simple to implement; the resulting square-root–style allocation is likely numerically stable.

- An attempt to combine first- and second-order information (Kronecker structure) in a single compensation procedure.

**Weaknesses:**

-  The key step \(\nabla L \approx \beta E\) (using residual \(E\) as a proxy gradient) lacks alignment evidence. There is no measurements of \(\cos(\nabla L, E)\), no bounds on \(\|\nabla L-\beta E\|\), and no layer-wise robustness to \(\beta\). With anisotropic curvature, \(E\) can point in low-salience directions, making compensation misaligned.
- Bit-allocation novelty is incremental. The closed-form rule reduces to classic sqrt/water-filling under convex sensitivity models; optimality is not shown jointly with codebook assignment and projection, so end-to-end optimality is unclear.
- Calibration regime likely underdetermined. Using \(\sim\)O(10^2) pairs for curvature/Kronecker stats in large VLMs is fragile; no curves vs. calibration size, no seed variance, and no distribution-shift tests.

**Questions:**

Do projection–compensation iterations show monotone decrease of a defined surrogate or residual norms? Any non-convergent layers?

---

### Official Review · Reviewer_ZaY5 · 2025-11-04

**Soundness:** 3
**Presentation:** 3
**Contribution:** 2
**Rating:** 4
**Confidence:** 5

**Summary:**

Authors propose a novel vector quantization (VQ) method for vision-language models (VLMs). Specifically, authors first analyze two fundamental challenges in applying VQ directly to VLMs. Ones is modality-induced weight heterogeneiity, another is error compensation mismatch from ignoring first-order gradients. To address these two challenges, authors propose their MSAVQ, which contains two main contributions: sensitivity-driven structure mixed-precision quantization strategy and gradient-aware error compensation. The proposed methods reveal their effectiveness on various popular VLMs and show appealing compression ratio.

**Strengths:**

1. Experiments are extensive.
2. The compression ratio is appealing, i.e., 2-bit quantization.
3. The experimental results show the effectiveness of the proposed methods.

**Weaknesses:**

1. In Line 50-51, authors claim that the memory usage of Qwen2-VL-72B exceeds the capacity of most edge devices during inference stage. However, the different model size of VLMs have already defined their deployment conditions. In other words, why should we apply such a huge model, like 72B VLMs, on edge devices? In my opinion, huge models deployed on cloud services, while tiny model, like qwen-0.6b (maybe with some distillation from huge models) can be deployed on edge devices for easy but fast inference.
2. The weight and token from which layer of which model in Figure 2 are plotted?  Also, how is the similarity computed, like the attention score after softmax? Lack necessary description for clarity.
3. In Figure 3, the red line is hard to distinguish. And is there similar phenomenon happened in other layers of LLaVA-OV or other models?
4. Figure 6 is too naive to get enough information about how are the "CSA/MRSBP/OBA" worked. Authors need to redesign and enrich the figure about overview framework.
5. In Appendix A.4, authors claim that the task loss is depends on data and layer, however, the KL divergence between the output of the quantized models and full-precision counterparts is the also depends on data and layer.
6. Why SSMQ can solve the first challenge in VQ of VLMS, i.e. the " modality-induced weight heterogeneiity". Authors first claim two challenges in VQ of VLMS in the section of abstract and introduction, then claim that the crucial challenge is "how to allocate limited bit budgets" in section 4.1, which is conflict in writing.

**Questions:**

see weaknesses.

---

### Note · Program_Chairs · 2025-11-28
**Submission Desk Rejected by Program Chairs**

SAC and AC agree that this paper plagiarizes significantly from this May arxiv paper: https://arxiv.org/abs/2505.22988v1